behaviour, developmental biology

lingual echolocation, *Rousettus aegyptiacus*, ontogeny, active sensing, pup behaviour, development

**Author for correspondence:**
Grace C. Smarsh
e-mail: gcsmarsh@gmail.com

# Hearing, echolocation, and beam steering from day 0 in tongue-clicking bats

Grace C. Smarsh[1,4], Yifat Tarnovsky[1,2] and Yossi Yovel[1,3]

[1]School of Zoology, Faculty of Life Sciences, [2]School of Neurobiology, Biochemistry, and Biophysics, Faculty of Life Sciences, and [3]Sagol School of Neuroscience, Tel Aviv University, Tel Aviv, IL 6997801, Israel
[4]Department of Brain Sciences, Weizmann Institute of Science, Rehovot, IL 7610001, Israel

GCS, 0000-0003-2641-2575

Little is known about the ontogeny of lingual echolocation. We examined the echolocation development of *Rousettus aegyptiacus*, the Egyptian fruit bat, which uses rapid tongue movements to produce hyper-short clicks and steer the beam's direction. We recorded from day 0 to day 35 postbirth and assessed hearing and beam-steering abilities. On day 0, *R. aegyptiacus* pups emit isolation calls and hyper-short clicks in response to acoustic stimuli, demonstrating hearing. Auditory brainstem response recordings show that pups are sensitive to pure tones of the main hearing range of adult *Rousettus* and to brief clicks. Newborn pups produced clicks in the adult paired pattern and were able to use their tongues to steer the sonar beam. As they aged, pups produced click pairs faster, converging with adult intervals by age of first flights (7–8 weeks). In contrast with laryngeal bats, *Rousettus* echolocation frequency and duration are stable through to day 35, but shift by the time pups begin to fly, possibly owing to tongue-diet maturation effects. Furthermore, frequency and duration shift in the opposite direction of mammalian laryngeal vocalizations. *Rousettus* lingual echolocation thus appears to be a highly functional sensory system from birth and follows a different ontogeny from that of laryngeal bats.

## 1. Background

Bat echolocation is highly adaptive, requiring extensive motor control to dynamically adjust spectrotemporal properties and beam shape to navigate sensory-behavioural challenges [1–6]. The ontogeny of echolocation is thus of special interest. Numerous studies have focused on bats that use laryngeal-produced signals. Some bat species are capable of hearing from birth, largely demonstrated from physiological recordings in the auditory brainstem [7–9]. Across families, pups begin to hear by the first- or second-week postbirth and have changes in frequency sensitivity as they age [10–14]. Pups are not born with full echolocation capacity. Neonatal pups generally produce echolocation precursors which change dramatically during development, including adjusting in frequency, duration, bandwidth, number of harmonics, steepness of the signal, and emission rate [7,8,10,15–21]. Echolocation signals and emission rate is usually adult-like by the time pups have reached full flying capacity (e.g. [16–18,20]). However, little is known regarding the developmental processes of click-based echolocation sensory systems. In this study, we examined the ontogeny of echolocation clicks in the Egyptian fruit bat, *Rousettus aegyptiacus*.

Click-based echolocation is found in several types of animals, including toothed whales, oilbirds and swiftlets, tenrecs, humans, and bats of the genus *Rousettus*, produced using different mechanisms [22–26]. *Rousettus* bats of the Family Pteropodidae produce clicks naturally using the tongue, unlike humans who must learn the technique. Rapid tongue movements produce hyper-short, broadband clicks that are steered in optimal directions for object targeting [2,27–31]. Beam forming and steering is crucial for many sensory systems, allowing individuals to focus spatially on relevant information in the

environment and disregard clutter. Laryngeal echolocating bats adjust beam shape and direction using various means including: adjusting signal frequency, mouth-gape, position of nasal structures, and head movements [3,32–35]. *Rousettus aegyptiacus* adults produce clicks in pairs, rapidly shifting the beam direction of each click within the pair without head movements. This beam steering is achieved by very rapid tongue movements which drive the beam direction in a left–right–right–left alternating pattern. In this specialized strategy, Egyptian fruit bats accurately point the maximum derivative of their beam towards objects of interest to improve its localization [28,36,37]. Because of the importance of beam steering for the success of this sensory system, in this study, we examined the echolocation emission patterns and the ability to steer the echolocation beam in young Egyptian fruit bat pups.

We assessed the ontogeny of lingual echolocation in Egyptian fruit bats by analysing click frequency, duration and temporal emission patterns recorded from pups once a week from day 0 to day 35 postbirth. Because of the different mechanism of signal production and strategy of echolocation use, we hypothesize that the ontogenetic patterns of lingual echolocation in *Rousettus* bats differ from that of laryngeal bats, and mammalian laryngeal vocalizations in general. We tested hearing abilities of newborn pups through both behavioural assessment and auditory brainstem response (ABR) recordings in response to sound, and determined the ability of newborn pups to steer the click beams with recordings from a two-microphone array. We expected that beam-steering abilities require experience to perform and will not be present in young pups.

## 2. Methods

### (a) Subject details

We captured pregnant *R. aegyptiacus* from a cave in Herzliya, Israel in 2017 and 2019 ($n = 19$, December 2017; $n = 15$, January and February 2019). We housed the bats in a colony room at the Zoological Gardens at Tel Aviv University on a reverse 12 h light cycle. The colony was maintained at a temperature between 25 and 28°C, and were fed a mixture of fresh fruits daily. We applied individual markings on the mothers' fur using hair bleach. Pregnancy was checked with an ultrasound. From mid-March to mid-April, we monitored the bats for new births daily. We began trials on day 0 postbirth.

### (b) Experimental protocol

#### (i) Hearing test

Pups of *R. aegyptiacus* are robust compared to many insectivorous species—they weigh approximately 17 to 24 g, are born with eyes open or closed, and have bent ear tips. During the first three weeks postbirth, pups are constantly attached to their mothers, including during foraging [38,39]. Like many bat species, if a pup becomes separated from its mother the pup produces loud, low-frequency isolation calls such that the mother can find and retrieve its offspring [40]. Behavioural audiograms use behavioural responses to assess hearing thresholds and can provide greater sensitivity than ABR alone [41–44]; however, behavioural audiograms require many trials in trained adults and are thus difficult to complete in pups. Nonetheless, using the vocal behaviour of Egyptian fruit bat pups, it is possible to design an appropriate test to determine the presence of hearing.

We assessed hearing presence by playing back noise and adult communication calls and assessing pup vocal responses.

In a normal colony, pups are regularly exposed to loud, low-frequency, adult social calls [45]. We conducted these trials in March and April, 2018. The calls were recorded from different adults previously [46]. Noise was flat-topped between 5 and 110 kHz, generated in Avisoft SasLab Lite. Each call file included an adult call repeated twice. Because *Rousettus* calls are multi-syllabic [46], intersyllable and intercall intervals varied across files. Accordingly, we modified the noise files to consist of noise bursts and silent intervals varying between 0.1 and 15 s. The total length of stimulus files varied between approximately 2.5 and 5.0 s. We created a total of 42 noise files and 87 call files. We calibrated noise and call stimuli to a peak amplitude of 78–80 dB SPL with a GRAS microphone (40DP 1/8″) and pre-amplifier (type 26AC) placed within the experimental box at the level of the pup's head. Because the microphone frequency response is flat and calibrated, it accounts for the entire frequency response of the playback system and speaker. On day 0, we detached the pups from the mother one at a time and placed with feet hanging on plastic mesh in a three-sided wooden box ($37 \times 40 \times 40$ cm) lined with sound-absorbing foam and a heating pad. Pups called spontaneously when first removed from the mother, but rapidly decreased in call rate ceasing usually 5 to 10 min after removal. To observe a clear response, we waited for the pup to stop calling for 30 s before starting the first stimulus. We played five different noise stimuli and five different call stimuli with 30 s of silence in between stimuli within sets, and 1 min of silence between noise and call sets. We randomized the order of the noise and call sets. We played the stimuli using the Avisoft Recorder software on a laptop connected to an Avisoft 116 Player and Vifa speaker, which was placed 20 cm away from the box at mid-height. We acoustically recorded the trial with a 116hm Avisoft Recorder (250 kHz sample rate) and Knowles FG microphone (10–120 kHz; 500 mV Pa$^{-1}$), and video recorded with a Sony video camera illuminated with a red light. We returned the pup to the mother approximately 30–40 min after detachment.

To assess pup sensitivity to different frequencies, we recorded ABRs of several day 0–2 pups. We sedated pups with a small dose of Midazolam ($1.0$ mg kg$^{-1}$) injected subcutaneously [47] and applied topical lidocaine (2%) to the sites of subdermal needle entry on the head for subcutaneous electrode recordings. Following Taiber *et al.* [48], we recorded ABRs in an acoustic chamber (MAC-1, Industrial Acoustic Company, Naperville, IL, USA). We used a calibrated set-up including an RZ6 multiprocessor, an MF1 speaker (Tucker-Davis Technologies, Alachua, FL) and a calibration microphone (ACO Pacific, Belmont, CA), connected to the BioSigRZ software program (Tucker-Davis Technologies, Alachua, FL). We placed the pup on a heating pad within the chamber, which was maintained at 37°C.

We presented each pup with a 0.1 ms click stimulus (most energy from 0 to 10 kHz, sloping downwards to 50 kHz) and subsequently 1 ms tones of 6, 12, 18, 24, 30 and 35 kHz, from low to high intensity (20 dB SPL–90 dB SPL) in steps of 5 dB. This specially calibrated system does not play beyond 35 kHz owing to limitations of the MF-1 speaker; however, this set of frequencies encompasses the main hearing region of adult *R. aegyptiacus* [41]. For each frequency-intensity combination, the pure tones were played 512 times and the responses were averaged to generate the overall 10 ms response signal which was displayed in the BioSigRZ program. We repeated this at least two times per frequency-intensity recorded, focusing on intensities at least three above and three below the suspected threshold to minimize experimental time. We ran additional repeats if the signal was noisy or unclear near a suspected threshold. Upon completion of the recordings, we injected a small dose of the antagonist Flumazenil (0.02 mg kg$^{-1}$ [47]) to awaken the pup if we did not observe sucking in response to stimulation after several minutes. Following a sucking response,

we placed the pup on the mother, provided her fruit and juice ad libitum, and kept the pair for observation with a heating pad overnight.

### (ii) Echolocation signal collection

We recorded pups on days 0, 7, 14, 21, 28 and 35 in a soundproofed flight room in March and April, 2019 (with the exception of one individual recorded 1 day late). We used a GRAS microphone and preamplifier (see specifications above) connected to a 116hm Avisoft Recorder and laptop with the Recorder software. The microphone was placed on a 1 m high platform with the pup held stable in the hand approximately 30 cm away. Recordings were collected continuously at 1 min intervals with a 250 kHz sample rate. We were not always able to collect clear recordings from the same pups on all days. We analysed recordings from 6 to 10 pups per day. We also recorded from five pups at the time they began to fly (generally between 7 and 8 weeks postbirth), to assess maturation effects at this stage. We released the pups from chest height five times in a row to confirm that the pup had reached minimum flying ability (fly while maintaining height).

We compared the pup echolocation signals to adult echolocation signals. We analysed intra- and interpair intervals ($n = 4$), frequency and duration from sets of echolocation from recordings of stationary (perched) adult *R. aegyptiacus* ($n = 3$) to compare with the pre-volant pups. We recorded adults in the same set-up as the pups at a 500 kHz sample rate. We held the adults by the lower body while allowing them to flap their wings in short bursts in the direction of the microphone, stimulating echolocation production. These adults were from a separate housed colony in the Zoological Garden.

### (iii) Beam recordings

To monitor beam steering, we used two Avisoft Knowles FG microphones with similar gains (0.17 V difference) connected to a 12-channel 1216hm Avisoft recorder. We placed the microphones on 1 m high platforms, angled at $45^{\circ}$ equidistant to the pup (66 cm away). Multichannel recordings were collected simultaneously with a 500 kHz sample rate. Microphones were placed in the same position during each recording session for consistency. We video recorded the sessions with a Sony video camera illuminated with red light. The video camera was placed at approximately 1 m height 50 cm across from the handler. We held the pup in the hand facing the video camera with body and head stable while stimulating the pup to echolocate. We assessed beaming ability for five pups (day 0–2 postbirth). Following all recordings and trials, we took weight and forearm measurement of the pup and weight of the mother, inspected the pair for overall good health and rewarded the mother with mango juice before returning them to the colony.

## (c) Quantification and statistical analysis

### (i) Hearing assessment

We analysed an equal number of clean trials in which noise was presented first and calls were presented first for 10 pups. From the oscillogram in Avisoft SASLab Lite, we quantified the number of isolation calls as well as echolocation clicks the pup produced during the 10 s prior to the start of the stimulus, and 10 s from the start of the stimulus. We calculated the average number of calls/clicks produced during silence pre-noise, noise, silence pre-call, and call for each of the 10 pups. We analysed the significance of response using repeated measures ANOVA and post hoc *t*-tests using the Real Statistics Resource Pack (v. 7.6) [49].

We plotted the ABR repeats and overall average for each frequency-intensity combination in MATLAB (R2020b). We defined the ABR threshold for each frequency as the lowest sound intensity at which a reproducible waveform was observable for the repetitions of the responses (electronic supplementary material, figure S1), as determined by two observers. We ensured that there were no more than 5 dB differences between the determined thresholds of the observers. For comparison to pups, we plotted the thresholds for five adult *R. aegyptiacus* whose ABRs were previously collected in the laboratory. There were only two instances in which the thresholds determined by the two observers differed by 10 dB. Those thresholds were determined as the intermediate amplitude.

### (ii) Processing and analysis of clicks

We analysed three temporal parameters: the click duration (duration), the interval within click pairs (intrapair interval) and the interval between click pairs (interpair interval); and the instantaneous frequency at the peak of the clicks. To measure intra- and interpair intervals, we selected high-quality sections of recordings in which there were four to eight pairs of clicks in a row uninterrupted by isolation calls. We analysed one–two sets of pulses per bat per day. We took measurements manually in Avisoft SasLab Lite by measuring start to start of clicks for intrapair interval, and midpair to midpair for interpair interval. For duration and instantaneous frequency measurements, we used SasLab Lite to target high signal-to-noise ratio sections (above approximately 60%) in the recordings to extract clicks for duration measurements. Because of the small number of points in the clicks, Fourier-based analyses are less accurate, so we used instantaneous frequency-based measurements which have been shown to be correlated to Fourier-based measurements [50]. In MATLAB (v. R2018b), we wrote a script targeting the peak of the envelope of the click and extracting the start and end times of the signal by measuring the points at −10 dB on either side of the peak. The script extracted the frequency of the wavelet by computing the instantaneous frequency of the signal and targeting the frequency at the point of peak amplitude. The script did not perform well on clicks with reverberation, which we excluded from analysis. We used the duration and instantaneous frequency measurements from at least 10 clear clicks per pup per day ($12 \pm 0.4$ clicks) for analysis.

We analysed the results using summary statistics with Real Statistics. Because not all parameters were normal nor were all pups recorded successfully on all days, we used general linear mixed models with day as a fixed factor and day | batID as random slope/random intercepts in MATLAB (v. R2018b), to analyse the effect of day on separate echolocation parameters. Adult signal parameters were extracted in the same method as the pups. We only used clear clicks approximately 0.5 V and higher ($\bar{x} = 0.7 \pm 0.2$ V, $n = 37$ clicks) for each adult from which to calculate individual means. We compared pup versus adult means on each day postbirth using Welch's *t*-test [51].

### (iii) Beaming analysis

To analyse echolocation in the beaming set-up, we first targeted sections of the recording sessions in which the pup was echolocating with head stable in the videos using MOVIEMAKER. We used BATALEF, an in-house MATLAB-based software, (e.g. [52]) to analyse pulses in associated multichannel recordings. We extracted amplitude measurements of simultaneous pulses in the left and right channel oscillograms using the automated peak to peak amplitude function. To compensate for the 0.17 V gain difference between microphones, we added 0.17 V to the click amplitudes of the lower amplitude channel. These amplitudes were compared across channels for each click to determine the directionality. We summed the number of correct left–right beam direction changes within each set of sequential pulses per pup and calculated the average proportion of correct transitions for all pups. We tested the significance of this proportion from random using a binomial exact test.

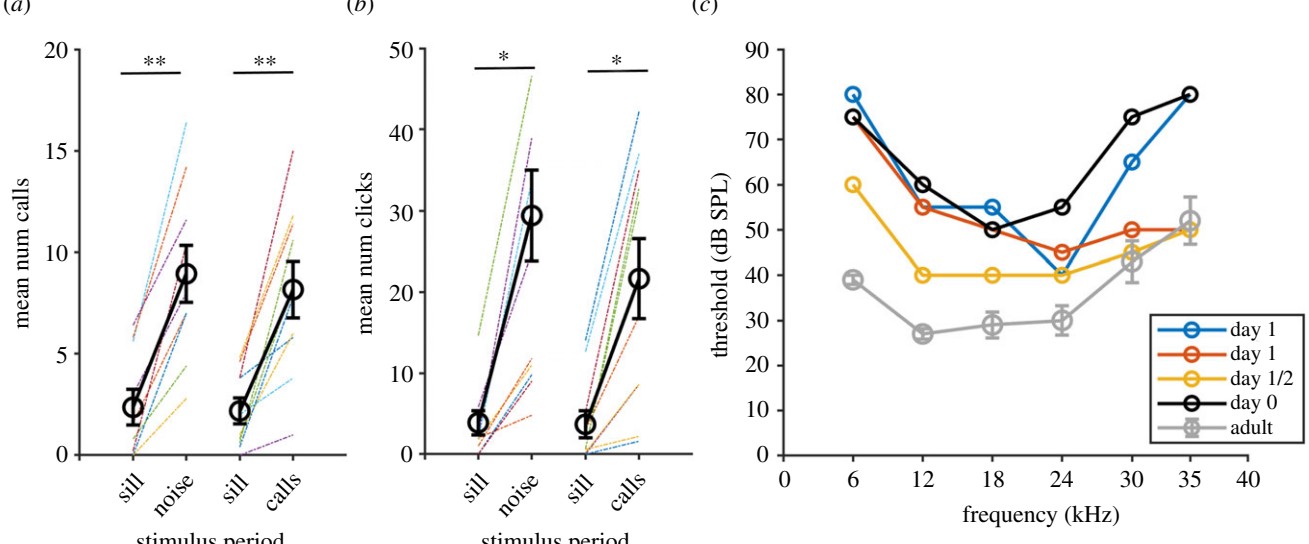

**Figure 1.** Hearing response of newborn pups from behavioural assessment (*a,b*) and auditory brainstem responses (*c*). Pup call response to silence and noise, and silence and calls is represented as the black lines (overall mean + s.e.) on (*a,b*). Coloured lines represent individuals. Audiogram plot (*c*) includes the mean + s.e. of thresholds for five adult bats and four pups (see legend for age). *$p < 0.01$; **$p < 0.001$. (Online version in colour.)

## 3. Results

### (a) Egyptian fruit bats hear and produce clicks from birth

Newborn pups produced echolocation clicks and social isolation calls readily when detached from the mother and in response to various sounds (figures 1 and 2*a*). day 0 pups produced significantly more isolation calls in response to playbacks of noise and adult communication calls compared to silence, demonstrating their ability to hear (figure 1*a*; mean ± s.e. hereon out; silence versus noise versus silence versus calls: $f_6 = 28.4$, $p < 0.0001$, repeated measures ANOVA; noise versus silence: $\bar{x}_{silence} = 2.36 \pm 0.88$ calls, $\bar{x}_{stimulus} = 8.94 \pm 1.41$ calls, $t_9 = -7.8$, $p < 0.0001$; calls versus silence: $\bar{x}_{silence} = 2.18 \pm 0.63$ calls, $\bar{x}_{stimulus} = 8.16 \pm 1.39$ calls, $t = -5.4$, $p < 0.001$; noise versus calls: $t = 1.22$, $p = 0.25$; $n = 10$; post hoc paired $t$-tests). Furthermore, pups significantly produced more echolocation clicks in response to playback of both noise and adult communication calls compared to silence (figure 1*b*; silence pre-noise versus noise versus silence pre-calls versus calls $f_6 = 6.56$, $p < 0.0001$, repeated measures ANOVA; noise versus silence pre-noise: $\bar{x}_{silence} = 3.88 \pm 1.51$ clicks, $\bar{x}_{stimulus} = 24.4 \pm 5.59$ clicks, $t = 4.52$, $p < 0.01$; calls versus silence pre-calls: $\bar{x}_{silence} = 3.68 \pm 1.69$ calls, $\bar{x}_{stimulus} = 21.62 \pm 4.94$ clicks, $t = 4.58$, $p < 0.01$; $n = 10$, post hoc paired $t$-tests). See the electronic supplementary material, Media and figure S2 for video and spectral examples of responses.

We recorded ABRs from four newborn pups (figure 1*c*; electronic supplementary material, figure S1). One was tested on day 0 postbirth, two on day 1, and one was either day 1 or 2 as it was born at the weekend. ABRs were collected for all pups in response to the click and all six pure tones. Pup sensitivity varied across individuals, but generally followed a somewhat convex shape. Adult curves were convex, with higher thresholds at 6, 30 and 35 kHz (figure 1). Pups generally had higher thresholds than adults, particularly at 6 kHz. At 35 kHz, two pups had the same thresholds as the adults at 50 db SPL, and two had far higher thresholds of 80 db SPL. Pup threshold to the click varied between 60–70 db SPL,

whereas the adults were more sensitive with threshold levels varying from 40 to 50 db SPL.

### (b) Pups can steer the echolocation beam from day 0

We assessed the beam-steering ability of newborn pups ($n = 3$ day 0, $n = 1$ day 1 and $n = 1$ day 2 postbirth) by analysing the relative amplitude of sequences of pulses recorded by two microphones placed equidistant from perched pups (see Methods). Newborn pups consistently shifted the echolocation beam between the left and right microphone. Pups mostly followed the adult left–right–right–left pattern of beaming that adults use regularly in flight, significantly transitioning to the correct direction approximately 60% of the time (figure 2*b*; 59.2 ± .05%, $n_{transitions} = 220$, $p < 0.01$, binomial exact test).

### (c) *Rousettus* pups exhibit innate temporal echolocation patterns

Already on day 0, pups produced clicks in the adult pattern with paired pulses separated by longer intervals between pairs (figure 2*b*). Intrapair intervals were slightly longer than the adults', but this difference was not significant, demonstrating the ability of fast tongue movements of newborn pups (figure 2*e* and table 1). Pups produced clicks in tighter pairs as they aged ($t_{504} = -3.016$, $p < 0.01$; generalized linear mixed model (GLMM)), with the same intrapair intervals as adults from day 14 on (figure 2*e* and table 1). The click-pair emission rate was slower on day 0, with interpair intervals significantly longer than adults (figure 2*f* and table 1). Pups produced pairs at a faster rate with shorter interpair intervals as they aged (figure 2*f* and table 1; $t_{426} = 6.68$, $p < 0.001$; GLMM), but interpair intervals were still significantly longer than adults at day 35 postbirth (figure 2*f* and table 1).

In laryngeal bats, emission rate changes have often been linked with flight development, which is achieved early (e.g. *Myotis lucificigus* day 17, *Myotis macrodactylus* day 19, *Noctilio albiventris* day 40) [16,17,20,53]. At day 35, *R. aegyptiacus* has

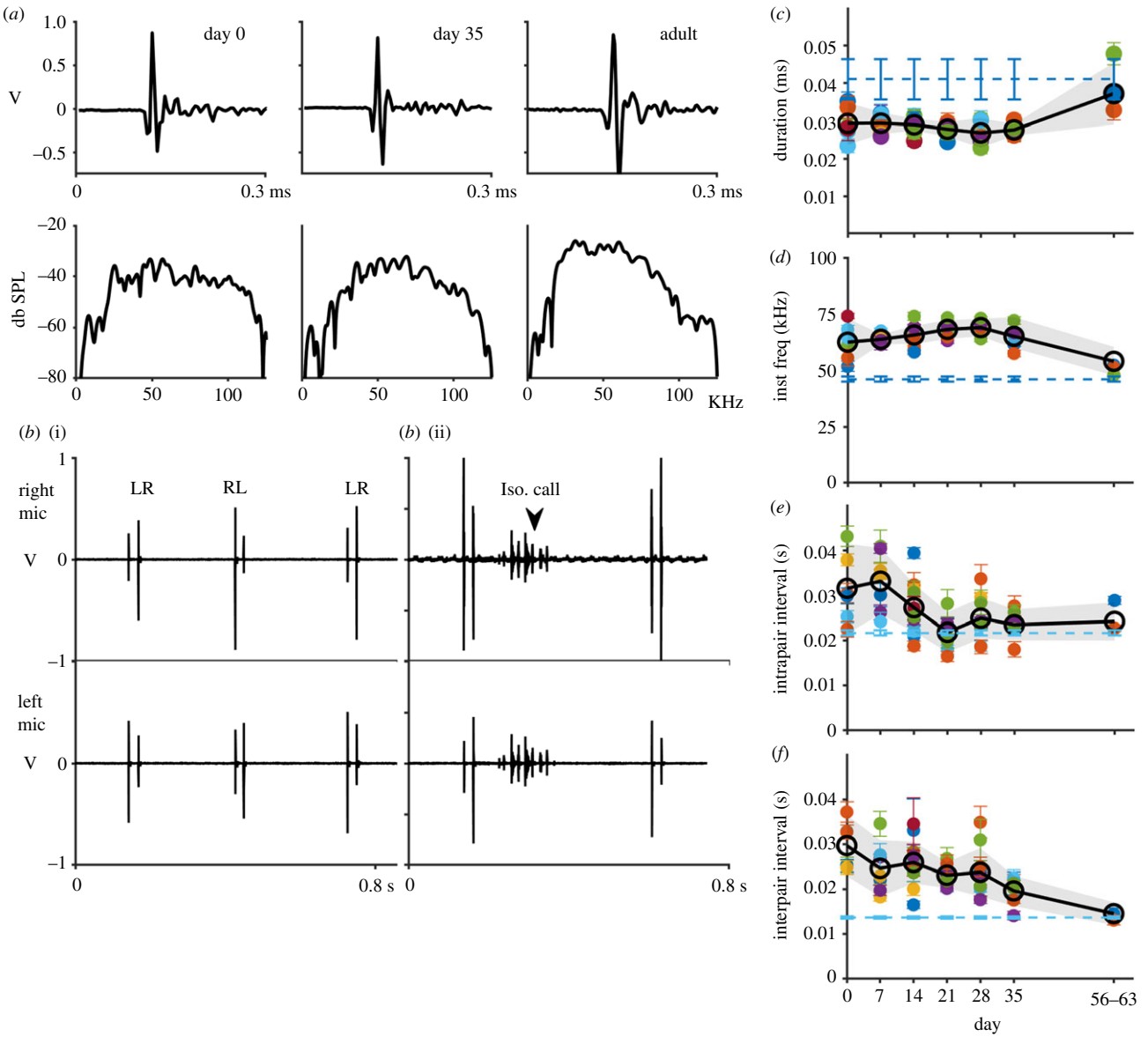

**Figure 2.** The ontogeny of Egyptian fruit bat echolocation. (*a*) Example echolocation clicks from a day 0 pup, day 35 pup and adult (dur = 24 μs) with power spectra below. (*b*(i)–(ii)) Two beam-steering examples were recorded from day 0 pups. Note the reverse intensity pattern in the right microphone (top) and the left microphone (bottom). Pups changed the beam direction in the right–left–left–right pattern. Example (*b*(ii)) includes an isolation call in between pairs. (*c*) Mean duration by day postbirth. (*d*) Mean instantaneous frequency by day postbirth. (*e*) Mean intrapair interval by day postbirth. (*f*) Mean interpair interval by day postbirth. (*c*–*f*) Black line is overall mean and grey shading represents 95% confidence intervals. Coloured circles are mean + s.e. of individuals. Dashed horizontal line shows the overall mean + s.e. for adults. (Online version in colour.)

not yet achieved flight. Although the mechanism of sound production differs, we hypothesized that interpair intervals become fully adult-like by the age of first flights with horizontal displacement, when sensing becomes crucial. To check this hypothesis, we recorded an additional five pups ($\bar{x}_{FA}$ = 69.9 mm) which were able to fly with at minimum horizontal displacement, which usually occurs at approximately 7–8 weeks of age. In support of the flight development hypothesis, both intra- and interpair intervals were similar to adults at age of first flights (figure 2*e*,*f*; intra: 0.028 ± 0.002 s; pup versus adult $t_{5.38}$ = 1.57, $p$ = 0.18; inter: 0.14 ± 0.009 s; pup versus adult $t_{4.89}$ = 0.91, $p$ = 0.41; Welch's *t*-test).

### (d) Signal properties change little during the first month

The echolocation clicks produced by day 0 pups resembled those of adults (figure 2*a*). Pup clicks were very short (36 μs

on average), but not significantly shorter than adults (figure 2*c* and table 1). Source levels were slightly lower than those reported for adults in the laboratory, ranging from 94.8 to 107 db SPL re 1 cm (reviewed in [27]). Pup clicks were significantly higher in frequency than adult clicks, as measured by the instantaneous frequency (figure 2*d* and table 1). Signal duration did not change through day 35, nor did the instantaneous frequency (figure 2*c*,*d* and table 1; dur: $t_{528}$ = −0.113, $p$ = 0.91; IF: $t_{528}$ = 0, $p$ = 0.99; GLMM). Click frequency and duration shifted significantly by the age of first flights (i.e. 7–8 weeks) (figure 2*c*,*d*; dur: 37.24 ± 2.29 μs; pup day 35 versus flight age, $t_{4.35}$ = 3.39, $p$ = 0.028; IF: 54.12 ± 2.29 kHz; pup day 35 versus flight age, $t_{8.45}$ = 3.86, $p$ < 0.01; Welch's *t*-test). The parameters shifted towards adult values, but frequency remained slightly higher in volant pups (dur pup versus adult: $t_{3.24}$ = 0.62, $p$ = 0.6; IF pup versus adult: $t_{5.63}$ = 3.06, $p$ = 0.03; Welch's *t*-test).

**Table 1.** Adult versus pup pup averages and Welch's *t*-test statistics per day (days 0–35 postbirth). (Note: adult frequency was higher than previously reported [27], owing to differences in microphone type and recording set-up (see Methods). Recordings were highly on-axis and in dose range with a sensitive microphone, resulting in higher instantaneous frequencies.)

| age | | adult | 0 | 7 | 14 | 21 | 28 | 35 |
|---|---|---|---|---|---|---|---|---|
| dur | $\bar{x} \pm$ s.e. μs | 41.1 ± 5.39 | 27.9 ± 1.96 | 28.74 ± 0.86 | 28.51 ± 0.73 | 26.6 ± 0.59 | 26.6 ± 1.24 | 26.9 ± 0.62 |
| | d.f. t p-value | | 2.5  2.29  0.15 | 2.1  2.25  0.15 | 2.07  2.3  0.15 | 2.05  2.66  0.12 | 2.21  2.61  0.12 | 2.05  2.6  0.12 |
| IF | $\bar{x} \pm$ s.e. kHz | 46.3 ± 0.12 | 62.5 ± 3.36 | 64.3 ± .62 | 65.9 ± 1.49 | 68.3 ± 1.41 | 68.8 ± 1.45 | 65.4 ± 1.83 |
| | d.f. t p-value | | 6.09  4.56  <0.01 | 3.17  13.49  <0.001 | 8.55  10.35  <0.001 | 7.0  11.9  <0.001 | 6.57  12.09  <0.001 | 7.91  8.8  <0.001 |
| intra int | $\bar{x} \pm$ s.e. s | 0.022 ± 0.0006 | .031. ± .003 | 0.033 ± .003 | 0.027 ± .002 | 0.021 ± .002 | 0.025 ± 0.002 | 0.023 ± .001 |
| | d.f. t p-value | | 6.32  2.033  0.09 | 8.23  3.08  0.015 | 11.98  1.128  0.28 | 8.88  1.48  0.17 | 10.69  0.14  0.89 | 7.82  0.71  0.49 |
| inter int | $\bar{x} \pm$ s.e. s | 0.14 ± 0.003 | 0.296 ± 0.021 | 0.246 ± 0.021 | 0.259 ± 0.017 | 0.231 ± 0.009 | 0.2376 ± 0.019 | 0.2 ± 0.011 |
| | d.f. t p-value | | 5.17  7.99  <0.001 | 6.21  5.746  0.001 | 9.47  7.8  0.001 | 6.97  10.67  <0.001 | 8.33  5.786  <0.001 | 6.71  6.38  <0.001 |

## 4. Discussion

The hearing abilities, signal design, emission temporal pattern and beam steering in newborn *Rousettus* suggest that many aspects of click-based echolocation are functional at birth. This is one of the earliest examples of active sensing control in an animal. In dolphins, clicks are produced only in the second week [54]. The sensory use of clicks by young *Rousettus* pups is unclear. The pre-volant pups emitted echolocation regularly in response to nearby sounds as early as day 0, and in a preliminary test with a week old pup the individual responded to echolocation passes as well (electronic supplementary material, figure S3). In laryngeal bats, the drastic adjustment of vocal signals in the first month corresponds with morphological changes supporting the superfast musculature of the larynx, auditory sensitivity and audiovocal feedback [8,10,14,53,55–59]. Laryngeal echolocating pups vary in their onset of hearing, but generally have lower auditory sensitivity than adults and hearing ranges restricted to lower frequencies (e.g. [7]). These high-frequency shifts correspond to maturation of the hair cells and to changes in selectivity in the auditory cortex [14,59]. In comparison to laryngeal echolocators, the hearing range of *Rousettus* is concentrated at lower frequencies with their most sensitive region between 8 and 30 kHz [41]. We found that *Rousettus* hearing is sensitive from birth to these frequencies although thresholds were higher than adults. Because adult *Rousettus* are less sensitive to high frequencies nor have an upwards shift in the frequency of the emitted click during development, we expect fewer adjustments of their auditory system compared to laryngeal pups (e.g. [11,47,60,61]). Indeed, prenatal work demonstrated that *Rousettus* pup cochlear growth rate occurs predominantly *in utero* but slows greatly postnatally compared to laryngeal bats [62], and that the developmental trajectory of ear features in lingual bats was more similar to non-echolocating mammals than to laryngeal bats [63].

Kössl *et al.* [59] found that certain neuronal circuits for echolocation ranging in the auditory cortex were already functional prior to the onset of echolocation in laryngeal pups. The ability of *Rousettus* pups to both perceive and acoustically respond to social calls and noise on day 0 shows that connections between audiocortical and motor circuitry are functional weeks before they need to perceive and interpret echoes in flight. We hypothesize that *Rousettus* pups respond with isolation calls and echolocation clicks to stimuli to attract their approaching mothers and perhaps also to sense their mothers (or other bats) approaching.

During development, laryngeal echolocation generally increases in frequency and decreases in duration, which is the opposite pattern of *Rousettus* clicks (e.g. [10,12,15,17–19]). We hypothesize that the production source (laryngeal versus lingual) underpins these differences. *Rousettus* produce the shortest clicks in animals except for Odontocetes, which have a specialized sound production mechanism [22,25]. Egyptian fruit bat tongues differ morphologically from those of insectivorous bats [64,65]. The bony, blood-innervated base and keratinized features of Egyptian fruit bat tongues may play a role in echolocation production [64], but currently, it is not clear which physiological and morphological adaptations allow the production of such brief clicks. For comparison, human tongue clicking is typically an order of magnitude longer [23]. *Rousettus aegyptiacus* tongue properties are not fully developed in newborns, particularly the corny

epithelium. Changes in tongue features probably correspond to the transition from a milk diet to a more abrasive, fruit-leaf-nectar diet [66]. The tongue's mechanical properties will effect acoustic sound production, and thus diet and corresponding morphological changes may have some affect on echolocation in young pups. The timing of flight development in pups corresponds with the beginning of pup weaning to a solid diet of fruits, and thus the echolocation maturation shift that we observed in 7- to 8-week-old pups may be a direct result of changes in tongue properties. The shift in click frequency and duration is unlikely to be related to changes in tongue size, as these parameters did not change during the first 35 days, but the pups grew linearly with age (figure 1$f,g$; day-FA linear regression: $y = 0.84x + 35$; $R^2 = 0.95$, $t = 39.56$, $p < 0.001$, $n_{pups} = 12$, $n_{points} = 72$). These parameters only weakly correlated with forearm length (dur-FA: $\rho = -0.35$, $p = 0.01$; IF-FA: $\rho = 0.38$, $p = 0.02$; $n_{points} = 41$).

Laryngeal bats increase pulse emission rate as they approach objects by producing pulses in pairs and then groups of pulses [4,67]. The ontogeny of this modulated emission rate is probably innate in laryngeal echolocating pups [68]. Similarly, *Rousettus* pups produced clicks in pairs in the adult pattern from birth. Adult *Rousettus* bats exhibit very little change in click rate in different tasks. Emission rate may increase in different light levels during the approach; however, the intrapair interval within click pairs does not change and the paired pattern is retained [27,69], rendering experience as pups begin to fly unnecessary for signal emission pattern. Our results further suggest that the neuromusculur innervation of the tongue must be established enough for pups to produce nearly adult-like clicks in pairs with very short intervals while controlling beam direction, indicating an earlier developmental process of the tongue than of the larynx. By comparison, *Rousettus* laryngeal-derived social calls undergo a vocal production learning process over the course of months [40,45]. In humans, tongue movements become faster and vowel sounds become shorter [70] as the jaw and tongue spatio-temporal coupling becomes tighter and more refined throughout childhood [71]. The increase in click-pair emission rate as the pups begin to fly may be linked to a greater synchronization of tongue musculature with respiration owing to higher-order neural processes, as has been demonstrated in fast-control laryngeal vocalizations [72].

## 5. Conclusion

Lingual and laryngeal echolocating bats share several ontogeny patterns, including early development of hearing, and pulse emission rate linked to flight development. However, this study points to a strong divergence in echolocation ontogeny in lingual bats. Pups have high vocal capabilities from birth, including the production of hyper-short, high-frequency clicks in the paired adult pattern, and impressively, the ability to steer the echolocation beam. Second, the echolocation spectrotemporal parameters do not change for the first month, and shift by flying age in the opposite direction of laryngeal bats and other mammals (frequency decreases and duration increase). The echolocation production mechanism probably underlies these key ontogenetic differences. Tongue maturation/diet effects or off-axis flight may underlie the changes in frequency and duration at flying age. Overall, the high echolocation capacity of neonates demonstrates that tongue morphology and neural innervation is well developed at birth. The ontogeny of active sensing abilities usually requires at least one to three weeks. In dolphins, for example, clear clicks are produced only in the second week [54]. Whisking in rats and discharge in electric fishes do not appear until the second-week postbirth [73,74]. We report a rare case in which most active sensing components are functional from day 0. Future work should focus on the ontogeny of the neural processing of the system, including the perception of echolocation at young ages, and beam-steering refinement during early experiences in navigation and foraging.

**Ethics.** This project was approved under IACUC guidelines, Animal Use Protocols no. 04-17-065, 04-18-030, 04-21-043.

**Data accessibility.** The datasets and code for this manuscript have been published on Zenodo: https://zenodo.org/record/5233673#.YXal-Jy8w2X0. Further information and requests for resources should be directed to and will be fulfilled by the Lead Contact, Dr Yossi Yovel (yossiyovel@gmail.com).

**Authors' contributions.** G.C.S.: conceptualization, data curation, formal analysis, funding acquisition, investigation, methodology, project administration, writing—original draft, writing—review and editing; Y.T.: data curation, formal analysis and methodology; Y.Y.: conceptualization, funding acquisition, methodology, resources, software, supervision, writing—review and editing. All authors gave final approval for publication and agreed to be held accountable for the work performed therein.

**Competing interests.** We declare we have no competing interests.

**Funding.** This research was supported by the Zuckerman STEM Leadership Program and partially by the European Research Council (grant no. ERC-GPSBAT).

**Acknowledgements.** We thank the reviewers for their time and detailed comments on the manuscript. We thank Prof. Karen Avraham for use of the ABR set-up, and Shahar Taiber for training on this set-up. We thank Dr Arjan Boonman's input on recording set-up and analysis, Dr Lee Harten's input on *Rousettus* husbandry and flight development, and Dr Maya Weinberg's veterinary advice. Dr Smadar Tal provided ultrasound expertise. We thank the volunteers and students who assisted with data collection and analysis: Rachael Cohen, Sara Diamond, Elia Yanko, Guy Gindis, Nati Pelag and Jacob Dembitzer.

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
