## [Peer Review File · Proceedings of the Royal Society B: Biological Sciences]

Review History

RSPB-2020-2971.R0 (Original submission)

Review form: Reviewer 1

Recommendation

Major revision is needed (please make suggestions in comments)

Scientific importance: Is the manuscript an original and important contribution to its field?

Acceptable

General interest: Is the paper of sufficient general interest?

Acceptable

Quality of the paper: Is the overall quality of the paper suitable?

Marginal

Is the length of the paper justified?

Yes

Should the paper be seen by a specialist statistical reviewer?

No

Do you have any concerns about statistical analyses in this paper? If so, please specify them explicitly in your report.

No

It is a condition of publication that authors make their supporting data, code and materials available - either as supplementary material or hosted in an external repository. Please rate, if applicable, the supporting data on the following criteria.

Is it accessible?

Yes

Is it clear?

Yes

Is it adequate?

Yes

Do you have any ethical concerns with this paper?

No

Comments to the Author

The study tests the ontogeny of echolocation in Rousettus bats by measuring hearing and quantifying echolocation click structure (temporal and spectral) and beam steering.

While I do believe that the overall findings are correct, I have issues with two of the methods described:

1) The hearing test is poorly justified, and not described in full detail in the methods. It is not until the results section you state that you use isolation call responses as an indication of hearing. How do you justify this method? From your description the pups will spontaneously produce the isolation calls, so a "response" cannot unequivocally be deemed an indication of hearing. Why not use a conventional method such as ABR?

I would also argue that to show a fully functional echolocation system, you need to test echolocation clicks and you need to measure the audiogram. Arguably, your response proxy could be an indication of hearing, but it is completely unquantifiable, the pup hearing could have an entirely different audiogram than the adult and still respond to the sounds you produce. Based on this, your conclusion about fully developed echolocation system at day 0 is not justified.

2) The beam steering test and comparison is too inaccurate to quantify the difference in aim between adults and pups - you use two different recordings setups (hand-held pups and perching adults), and your recording setup consists of only two microphones. Given that the animals can steer the beam and readily modify the separation between click-pairs, this method is much too imprecise to quantify this.

Line 22-23. This statement needs references.

Line 32-33. Your argument indicates that this study will reveal general information about click-based echolocation systems. Given the difference between the click-echolocation systems you list later (birds, humans and whales), I find this claim highly speculative. I would suggest rewording to indicate that this sentence only concerns Rousettus click-echolocation or to justify its general applicability.

Line 45-49. None of the references listed (11,25,26) are for the features of Rousettus echolocation you describe...

Line 82. You state that "Noise and call files were calibrated to a peak amplitude of 78-80 dB SPL". This is very ambiguous. Is this measured at the pup's location? If so, write it explicitly. Is the measurement performed in the box? What is the size of the box? The box will have a large impact

on the sound field experienced by the pup, as such the measurement should be performed in the box and it should be stated very clearly if you did so. Otherwise your claimed exposure level is not trustworthy. Also, did you account for the frequency response of the speaker in this calibration?

Line 75-96. How do you measure the hearing with this approach? I can see that you quantify responses from Line 200-201, but that should be stated here, otherwise the reader is left guessing at the method. How do you justify this method, is there a study that does so?

Line 100. Gras should be GRAS and you should state size (1/4", 1/8"?) and model.

Line 122. Did you not calibrate the Knowles microphones? Given your method, I appreciate that you do not need an absolute calibration of the entire frequency response, but a simple relative calibration should be performed (exposing both microphones to the same sound) to test how similar they are.

Line 121-134. I find this method highly inaccurate and artificial. You are holding the animal in your hand, why not place it on a platform or hang it from a mesh?

Line 171-173. I do not understand this sentence.

Line 180-182. You mention an "automated measurement feature" – you should either explain this in detail or post a reference that does so.

Line 187-193. You compare pup beam-alternations to adult beam-alternations using an intensity ratio. Given the difference in measurement scenario and low control you have over aim (hand-held pups compared to perching adults and only two microphones for animals that can vary the beam aim/beam separation) this method seems highly inaccurate. I agree that you can test if the left-left – right-right pattern is present, but comparing angles is in my opinion too uncertain with this method.

Line 196-206. This method is not justified in your description. Interpreting calling response as validation of hearing is not entirely unjustified, but it does not guarantee it either. It is also very questionable to test the ontogeny of echolocation without exposing the animals to echolocation sounds. While you may be covering the frequency range, the call duration is dramatically different. Again, an appropriate method would be ABR instead.

Line 246-257. It would be very informative to see a spectrum of the signals. While your argument concerning fft-points from line 156-157 is somewhat valid, there is a vast selection of toothed-whale literature on this that provides justification and the means to do so.

Review form: Reviewer 2

Recommendation

Accept with minor revision (please list in comments)

Scientific importance: Is the manuscript an original and important contribution to its field?

Good

General interest: Is the paper of sufficient general interest?

Excellent

Quality of the paper: Is the overall quality of the paper suitable?

Excellent

Is the length of the paper justified?

Yes

Should the paper be seen by a specialist statistical reviewer?

No

Do you have any concerns about statistical analyses in this paper? If so, please specify them explicitly in your report.

No

It is a condition of publication that authors make their supporting data, code and materials available - either as supplementary material or hosted in an external repository. Please rate, if applicable, the supporting data on the following criteria.

Is it accessible?

Yes

Is it clear?

Yes

Is it adequate?

Yes

Do you have any ethical concerns with this paper?

No

Comments to the Author

The ms by Smarsh and Yovel describes ontogenetic development of auditory responsiveness and of echolocation calls in the bat *Rousettus*. In most laryngeal bats, call production can be delayed after birth and the emission of adult-like calls often is coupled to the onset of flight. *Rousettus*, however, uses a simpler lingual echolocation production system that is based on tongue clicking. The authors show that right after birth, *Rousettus* echolocates with clicks and the clicks are quite adult like in terms of their pairwise production. In addition, the neonate bats emit left/right click amplitude patterns that are similar to beam steering of adults. In the next weeks of life, the interpair interintervalls grow successively shorter and the call duration increases while the instantaneous frequency is reduced somewhat. Those changes, however, are small, compared to more massive changes during the ontogeny of laryngeal bats. In addition, the authors showed that the young bats increase the number of their own emitted isolation calls when louder sounds like echolocation clicks are presented. This of course is not a demonstration of intact hearing, but at least shows that the bats are reactive to some sounds right after birth.

The short ms generally is written well and the methods are sound. The data are important when assessing evolution of echolocation and comparing laryngeal versus lingual sound production and therefore should be published in Proc B. Lingual sound production and a concomitant perception of the clicks seems to be an innate feature and is less shaped by experience, inner ear maturation, and sensitive periods, as it is the case in laryngeal sound production.

Specific comments

Title:

your study does not demonstrate intact adult-like hearing, you did not show hearing threshold curves. But you showed that neonates did respond to echolocation signals, please change the title accordingly.

Introduction (and discussion):

I am a little bit surprised that you are not aware of a series of relevant literature with ample data on postnatal maturation of echolocation calls and corresponding behavioural adaptations in laryngeal bats (Vater M Kössl M Foeller E Coro F Mora E Russell I (2003) Development of Echolocation Calls in the Mustached Bat, *Pteronotus parnellii*. *J Neurophysiol* 90: 2274–2290) as well as their corresponding cochlear (Kössl M Foeller E Drexler M Vater M Mora E Coro E Russell IJ (2003). Postnatal development of cochlear function in the mustached bat, *Pteronotus parnellii*. *J Neurophysiol* 90:2261-2273) and auditory system maturation (Vater M Foeller E Mora EC Coro F Russell IJ Kössl M (2010) Postnatal Maturation of Primary Auditory Cortex in the Mustached Bat, *Pteronotus parnellii* *J Neurophysiol* 103: 2339–2354). Those data very nicely show that the mustached bat, as laryngeal bat, takes long time for maturation and would strongly emphasize the point that you make in your ms. Please cite and discuss those data.

Experimental protocol:

Hearing test: To use echolocation signals for your auditory responsiveness test makes sense, but what was the rationale of using 5-110 kHz noise which probably extends beyond the auditory range of those bats. Are there hearing threshold curves obtained with pure tones available for young *Rousettus* (maybe from another publication)?

Give a little more information about the behaviour of the young bats, did they usually hang on their mothers or stayed very close? In which natural situation do they emit echolocation calls?

Echolocation signal collection: what was the sound pressure level of the echolocation calls of the young bats? Since their tongues still undergo anatomical maturation, were the signal levels lower than those from adults?

Hearing assessment: When you acoustically stimulated the bats and they produced more isolation calls, they also emitted echolocation calls in this situation. Was their number also increased?

Results

L199: detachment of pups from mothers also happens naturally? Stay the pups at the roost while the mothers fly out to feed?

Discussion

L278 I am not quite sure how to interpret your statement about cochlear growth rate. Do you want to say that cochlear development in laryngeal bats is finished earlier than cochlear development in laryngeal bats (which would support your suggestion that hearing also matures faster)? I would not rely too much on macroscopic cochlea growth data since there is an extended period of maturation of cochlear function after the cochlea has reached its final size. In this respect, microstructural changes of basilar membrane and tectorial membrane stiffness and coupling between hair cells and those membranes are important and will increase cochlear sensitivity in a frequency specific way in young bats. Could you give more information what you are referring to here?

Decision letter (RSPB-2020-2971.R0)

12-Jan-2021

Dear Dr Smarsh:

I am writing to inform you that your manuscript RSPB-2020-2971 entitled "Intact hearing, echolocation, and beam steering from Day 0 in tongue-clicking bats" has, in its current form, been rejected for publication in Proceedings B.

This action has been taken on the advice of referees, who have recommended that substantial revisions are necessary. With this in mind we would be happy to consider a resubmission, provided the comments of the referees are fully addressed. However please note that this is not a provisional acceptance.

Sincerely,
Dr Maurine Neiman
mailto: proceedingsb@royalsociety.org

Associate Editor
Comments to Author:

Thank you for submitting your manuscript to Proceedings B. We have now received two reviews of your manuscript. One of the reviewers agrees that this study supports most of your conclusions and is an important contribution to understanding the evolution and development of diverse types of echolocation. However, both reviewers express concerns about the methods used to assess hearing in the pups, and both object to the conclusion that pups have "intact" hearing as stated in the title. The second reviewer is also concerned that the method used to assess beam steering was not accurate enough to compare with data from adults, and therefore, combined with the concerns about hearing, the conclusion that the pups of this species are capable of echolocation from day zero has not been adequately demonstrated. Both reviewers provide valuable comments and suggestions that will improve a future version of the manuscript.

Reviewer(s)' Comments to Author:

Referee: 1

Comments to the Author(s)

The study tests the ontogeny of echolocation in Rousettus bats by measuring hearing and quantifying echolocation click structure (temporal and spectral) and beam steering.

While I do believe that the overall findings are correct, I have issues with two of the methods described:

1) The hearing test is poorly justified, and not described in full detail in the methods. It is not until the results section you state that you use isolation call responses as an indication of hearing. How do you justify this method? From your description the pups will spontaneously produce the isolation calls, so a “response” cannot unequivocally be deemed an indication of hearing. Why not use a conventional method such as ABR?

I would also argue that to show a fully functional echolocation system, you need to test echolocation clicks and you need to measure the audiogram. Arguably, your response proxy could be an indication of hearing, but it is completely unquantifiable, the pup hearing could have an entirely different audiogram than the adult and still respond to the sounds you produce. Based on this, your conclusion about fully developed echolocation system at day 0 is not justified.

2) The beam steering test and comparison is too inaccurate to quantify the difference in aim between adults and pups – you use two different recordings setups (hand-held pups and perching adults), and your recoding setup consists of only two microphones. Given that the animals can steer the beam and readily modify the separation between click-pairs, this method is much too imprecise to quantify this.

Line 22-23. This statement needs references.

Line 32-33. Your argument indicates that this study will reveal general information about click-based echolocation systems. Given the difference between the click-echolocation systems you list later (birds, humans and whales), I find this claim highly speculative. I would suggest rewording to indicate that this sentence only concerns *Rousettus* click-echolocation or to justify its general applicability.

Line 45-49. None of the references listed (11,25,26) are for the features of *Rousettus* echolocation you describe...

Line 82. You state that “Noise and call files were calibrated to a peak amplitude of 78-80 dB SPL”. This is very ambiguous. Is this measured at the pup’s location? If so, write it explicitly. Is the measurement performed in the box? What is the size of the box? The box will have a large impact on the sound field experienced by the pup, as such the measurement should be performed in the box and it should be stated very clearly if you did so. Otherwise your claimed exposure level is not trustworthy. Also, did you account for the frequency response of the speaker in this calibration?

Line 75-96. How do you measure the hearing with this approach? I can see that you quantify responses from Line 200-201, but that should be stated here, otherwise the reader is left guessing at the method. How do you justify this method, is there a study that does so?

Line 100. Gras should be GRAS and you should state size (1/4”, 1/8”) and model.

Line 122. Did you not calibrate the Knowles microphones? Given your method, I appreciate that you do not need an absolute calibration of the entire frequency response, but a simple relative calibration should be performed (exposing both microphones to the same sound) to test how similar they are.

Line 121-134. I find this method highly inaccurate and artificial. You are holding the animal in your hand, why not place it on a platform or hang it from a mesh?

Line 171-173. I do not understand this sentence.

Line 180-182. You mention an “automated measurement feature” – you should either explain this in detail or post a reference that does so.

Line 187-193. You compare pup beam-alternations to adult beam-alternations using an intensity ratio. Given the difference in measurement scenario and low control you have over aim (hand-held pups compared to perching adults and only two microphones for animals that can vary the beam aim/beam separation) this method seems highly inaccurate. I agree that you can test if the left-left – right-right pattern is present, but comparing angles is in my opinion too uncertain with this method.

Line 196-206. This method is not justified in your description. Interpreting calling response as validation of hearing is not entirely unjustified, but it does not guarantee it either. It is also very questionable to test the ontogeny of echolocation without exposing the animals to echolocation sounds. While you may be covering the frequency range, the call duration is dramatically different. Again, an appropriate method would be ABR instead.

Line 246-257. It would be very informative to see a spectrum of the signals. While your argument concerning fft-points from line 156-157 is somewhat valid, there is a vast selection of toothed-whale literature on this that provides justification and the means to do so.

Referee: 2

Comments to the Author(s)

The ms by Smarsh and Yovel describes ontogenetic development of auditory responsiveness and of echolocation calls in the bat *Rousettus*. In most laryngeal bats, call production can be delayed after birth and the emission of adult-like calls often is coupled to the onset of flight. *Rousettus*, however, uses a simpler lingual echolocation production system that is based on tongue clicking. The authors show that right after birth, *Rousettus* echolocates with clicks and the clicks are quite adult like in terms of their pairwise production. In addition, the neonate bats emit left/right click amplitude patterns that are similar to beam steering of adults. In the next weeks of life, the interpair interintervalls grow successively shorter and the call duration increases while the instantaneous frequency is reduced somewhat. Those changes, however, are small, compared to more massive changes during the ontogeny of laryngeal bats. In addition, the authors showed that the young bats increase the number of their own emitted isolation calls when louder sounds like echolocation clicks are presented. This of course is not a demonstration of intact hearing, but at least shows that the bats are reactive to some sounds right after birth.

The short ms generally is written well and the methods are sound. The data are important when assessing evolution of echolocation and comparing laryngeal versus lingual sound production and therefore should be published in Proc B. Lingual sound production and a concomitant perception of the clicks seems to be an innate feature and is less shaped by experience, inner ear maturation, and sensitive periods, as it is the case in laryngeal sound production.

Specific comments

Title:

your study does not demonstrate intact adult-like hearing, you did not show hearing threshold curves. But you showed that neonates did respond to echolocation signals, please change the title accordingly.

Introduction (and discussion):

I am a little bit surprised that you are not aware of a series of relevant literature with ample data on postnatal maturation of echolocation calls and corresponding behavioural adaptations in laryngeal bats (Vater M Kössl M Foeller E Coro F Mora E Russell I (2003) Development of Echolocation Calls in the Mustached Bat, *Pteronotus parnellii*. *J Neurophysiol* 90: 2274–2290) as well as their corresponding cochlear (Kössl M Foeller E Drexler M Vater M Mora E Coro E Russell IJ (2003). Postnatal development of cochlear function in the mustached bat, *Pteronotus parnellii*. *J Neurophysiol* 90:2261-2273) and auditory system maturation (Vater M Foeller E Mora EC Coro F Russell IJ Kössl M (2010) Postnatal Maturation of Primary Auditory Cortex in the Mustached Bat, *Pteronotus parnellii* *J Neurophysiol* 103: 2339-2354). Those data very nicely show that the

mustached bat, as laryngeal bat, takes long time for maturation and would strongly emphasize the point that you make in your ms. Please cite and discuss those data.

Experimental protocol:

Hearing test: To use echolocation signals for your auditory responsiveness test makes sense, but what was the rationale of using 5-110 kHz noise which probably extends beyond the auditory range of those bats. Are there hearing threshold curves obtained with pure tones available for young Rousettus (maybe from another publication)?

Give a little more information about the behaviour of the young bats, did they usually hang on their mothers or stayed very close? In which natural situation do they emit echolocation calls?

Echolocation signal collection: what was the sound pressure level of the echolocation calls of the young bats? Since their tongues still undergo anatomical maturation, were the signal levels lower than those from adults?

Hearing assessment: When you acoustically stimulated the bats and they produced more isolation calls, they also emitted echolocation calls in this situation. Was their number also increased?

Results

L199: detachment of pups from mothers also happens naturally? Stay the pups at the roost while the mothers fly out to feed?

Discussion

l.278 I am not quite sure how to interpret your statement about cochlear growth rate. Do you want to say that cochlear development in lingual bats is finished earlier than cochlear development in laryngeal bats (which would support your suggestion that hearing also matures faster)? I would not rely too much on macroscopic cochlea growth data since there is an extended period of maturation of cochlear function after the cochlea has reached its final size. In this respect, microstructural changes of basilar membrane and tectorial membrane stiffness and coupling between hair cells and those membranes are important and will increase cochlear sensitivity in a frequency specific way in young bats. Could you give more information what you are referring to here?

Author's Response to Decision Letter for (RSPB-2020-2971.R0)

See Appendix A.

RSPB-2021-1714.R0

Review form: Reviewer 1

Recommendation

Accept as is

Scientific importance: Is the manuscript an original and important contribution to its field?

Good

General interest: Is the paper of sufficient general interest?

Acceptable

Quality of the paper: Is the overall quality of the paper suitable?

Good

Is the length of the paper justified?

Yes

Should the paper be seen by a specialist statistical reviewer?

No

Do you have any concerns about statistical analyses in this paper? If so, please specify them explicitly in your report.

No

It is a condition of publication that authors make their supporting data, code and materials available - either as supplementary material or hosted in an external repository. Please rate, if applicable, the supporting data on the following criteria.

Is it accessible?

Yes

Is it clear?

Yes

Is it adequate?

Yes

Do you have any ethical concerns with this paper?

No

Comments to the Author

The authors have addressed all my concerns in a very nice way and I have no further comments. I am happy to recommend the study for publication

Decision letter (RSPB-2021-1714.R0)

28-Sep-2021

Dear Dr Smarsh

I am pleased to inform you that your manuscript RSPB-2021-1714 entitled "Hearing, echolocation, and beam steering from Day 0 in tongue-clicking bats" has been accepted for publication in Proceedings B.

The referee(s) have recommended publication, but also suggest some minor revisions to your manuscript. Therefore, I invite you to respond to the referee(s)' comments and revise your manuscript. Because the schedule for publication is very tight, it is a condition of publication that you submit the revised version of your manuscript within 7 days. If you do not think you will be able to meet this date please let us know.

NB. From April 1 2013, peer reviewed articles based on research funded wholly or partly by RCUK must include, if applicable, a statement on how the underlying research materials – such

as data, samples or models – can be accessed. This statement should be included in the data accessibility section.

[http://datadryad.org/submit?journalID=RSPB&manu=\(Document not available\)](http://datadryad.org/submit?journalID=RSPB&manu=(Document%20not%20available)) which will take you to your unique entry in the Dryad repository. If you have already submitted your data to dryad you can make any necessary revisions to your dataset by following the above link. Please see <https://royalsociety.org/journals/ethics-policies/data-sharing-mining/> for more details.

Sincerely,
Dr Maurine Neiman
<mailto:proceedingsb@royalsociety.org>

Associate Editor
Board Member
Comments to Author:
Thank you for your careful revisions of this manuscript.

Reviewer(s)' Comments to Author:

Referee: 1

Comments to the Author(s).

The authors have addressed all my concerns in a very nice way and I have no further comments. I am happy to recommend the study for publication

Author's Response to Decision Letter for (RSPB-2021-1714.R0)

See Appendix B.

Decision letter (RSPB-2021-1714.R1)

06-Oct-2021

Dear Dr Smarsh

I am pleased to inform you that your manuscript entitled "Hearing, echolocation, and beam steering from Day 0 in tongue-clicking bats" has been accepted for publication in Proceedings B.

Data Accessibility section

Open Access

Paper charges

Sincerely,

Appendix A

12-Jan-2021

Dear Dr Smarsh:

I am writing to inform you that your manuscript RSPB-2020-2971 entitled "Intact hearing, echolocation, and beam steering from Day 0 in tongue-clicking bats" has, in its current form, been rejected for publication in Proceedings B.

This action has been taken on the advice of referees, who have recommended that substantial revisions are necessary. With this in mind we would be happy to consider a resubmission, provided the comments of the referees are fully addressed. However please note that this is not a provisional acceptance.

Sincerely,

Dr Maurine Neiman
mailto: proceedingsb@royalsociety.org

Associate Editor
Comments to Author:

Thank you for submitting your manuscript to Proceedings B. We have now received two reviews of your manuscript. One of the reviewers agrees that this study supports most of your conclusions and is an important contribution to understanding the evolution and development of diverse types of echolocation. However, both reviewers express concerns about the methods used to assess hearing in the pups, and both object to the conclusion that pups have “intact” hearing as stated in the title. The second reviewer is also concerned that the method used to assess beam steering was not accurate enough to compare with data from adults, and therefore, combined with the concerns about hearing, the conclusion that the pups of this species are capable of echolocation from day zero has not been adequately demonstrated. Both reviewers provide valuable comments and suggestions that will improve a future version of the manuscript.

Thank you. We have addressed the concerns regarding hearing in detail, and have collected ABRs in response to pure tones and a click to better address the concerns regarding hearing abilities in newborn pups. Hearing is not as sensitive as adults on Day 0 but the hearing range covers the main range for *Rousettus*. While not fully intact, the results point to another difference in ontogeny between laryngeal and lingual echolocating bats, in which laryngeal bats generally do not hear the main hearing range at this age and have an upward shift in frequency during development.

Regarding the behavioral tests on Day 0, we have added an additional analysis and figure of the click response to stimuli on Day 0, to provide greater information on echolocation use in newborns, as requested by Reviewer 2.

We have revisited the beam steering concerns as well, and agree with the reviewer’s argument that we cannot compare beam angle at this time. We have removed this, but have kept the assessment of directional changes, with which neither reviewers had faults.

We have added additional figures of the power spectra of the clicks, and added supplemental figures and a video to help illustrate the behavior and vocal abilities of young pups.

Below we have sought to fully address each reviewers’ individual comments and adjust the manuscript accordingly.

Reviewer(s)' Comments to Author:

Referee: 1

Comments to the Author(s)

The study tests the ontogeny of echolocation in *Rousettus* bats by measuring hearing and quantifying echolocation click structure (temporal and spectral) and beam steering. While I do believe that the overall findings are correct, I have issues with two of the methods described:

1) The hearing test is poorly justified, and not described in full detail in the methods. It is not until the results section you state that you use isolation call responses as an indication of hearing. How do you justify this method? From your description the pups

will spontaneously produce the isolation calls, so a “response” cannot unequivocally be deemed an indication of hearing. Why not use a conventional method such as ABR? I would also argue that to show a fully functional echolocation system, you need to test echolocation clicks and you need to measure the audiogram. Arguably, your response proxy could be an indication of hearing, but it is completely unquantifiable, the pup hearing could have an entirely different audiogram than the adult and still respond to the sounds you produce. Based on this, your conclusion about fully developed echolocation system at day 0 is not justified.

Thank you for your careful revision of our manuscript and detailed comments. We have sought to address your concerns in detail below and adjust accordingly. First, here we will address in detail the concerns about testing hearing, as this came up several times in your comments:

We see and agree with your point that our conclusion on fully intact hearing could not be justified as we did not show the sensitivity of hearing to different frequencies. It is important to note that behavior has been and is still used to assess hearing abilities. Behavioral audiograms can also be considered a “conventional” method as they are widely used across taxa and have been conducted successfully in bats for decades (e.g. Dalland 1965, Long & Schnitzler 1975, Suthers & Summers 1980, Schmidt et al 1984, Esser & Schmidt 1990) and are argued to demonstrate greater sensitivity than ABRs (e.g. Wenstrup 1984, Obrist & Wenstrup 1998, Koay et al. 1998, Lattenkamp et al 2020).

Our initial interest was presence of hearing. This can be assessed behaviorally by a significant response to acoustic stimuli. The design is similar to numerous other acoustic playback studies in bats, birds, and other taxa which establish recognition, discrimination, or detection of sounds with a significant behavioral response. We have reviewed and added/adjusted the methods and results regarding this test, and additionally analyzed the bats’ response to sound using echolocation, which is also significant (L 80-104, 166-172, 227-241, Fig 1, Supplementary Fig 1). To further illustrate the responsiveness of these robust animals, I have included an example video clip from one of our trials, as well as spectrograms in the supplements.

Moreover, in order to address the reviewer’s concerns regarding frequency sensitivity we recorded ABRs for four pups Day 0 to Day 2 in response to tone pips (6, 12, 18, 24, 30, 35 KHz) and a 0.1 ms click (0 to 50 kHz, most energy 0 to 10 kHz). These frequencies encompass the most sensitive hearing range of *R. aegyptiacus* (8 to 25 kHz, Koay et al. 1998). This part of the project was conducted with Yifat Tarnovsky, a PhD candidate in the Yovel lab, whose has training and experience in recording ABRs in *R. aegyptiacus*. Additionally, we included adult thresholds acquired from ABRs that Tarnovsky previously recorded in the same setup (L 115-128, 173-181, 242-250, Fig 1).

2) The beam steering test and comparison is too inaccurate to quantify the difference in aim between adults and pups – you use two different recordings setups (hand-held pups and perching adults), and your recording setup consists of only two microphones. Given that the animals can steer the beam and readily modify the separation between click-pairs, this method is much too imprecise to quantify this.

We accept this comment and removed this assessment of aim comparison.

Line 22-23. This statement needs references.

We have added references

Line 32-33. Your argument indicates that this study will reveal general information about click-based echolocation systems. Given the difference between the click-echolocation systems you list later (birds, humans and whales), I find this claim highly speculative. I would suggest rewording to indicate that this sentence only concerns *Rousettus* click-echolocation or to justify its general applicability.

We did not intend to make this claim, so we have reformulated this section.

We provided context regarding click echolocation systems as it is not specific to bats and there is much to learn about click-based sensory systems (L 30, L32-33). On L 31 we specifically state that we investigate the *Rousettus* click sensory system. On lines 34-35 we now state that the mechanism of click based echolocation differ, and then go on to discuss lingual echolocation systems specifically. We hope this is clear now

Line 45-49. None of the references listed (11,25,26) are for the features of *Rousettus* echolocation you describe...

Thank you for this catch. We have corrected this reference library issue and checked the rest of the document to ensure there were no others.

Line 82. You state that "Noise and call files were calibrated to a peak amplitude of 78-80 dB SPL". This is very ambiguous. Is this measured at the pup's location? If so, write it explicitly. Is the measurement performed in the box? What is the size of the box? The box will have a large impact on the sound field experienced by the pup, as such the measurement should be performed in the box and it should be stated very clearly if you did so. Otherwise your claimed exposure level is not trustworthy. Also, did you account for the frequency response of the speaker in this calibration?

Yes, this is important to include. We did calibrate the stimuli by holding the calibrated GRAS microphone from within the box at the location of the pup's head when the pup is perching there. We have added this point here L 88-L 90.

We have added the details that the box was lined with foam to reduce echoes, and its dimensions on L 93-94.

We considered the frequency response of the speaker before using it for the experiment. It covers 1-120 kHz, with variability of up to 6 db between 1 and 60 kHz. Since the response of the microphone we used to calibrate the system is flat and calibrated, it accounts for the entire frequency response of the system including the speaker. We have added this point here L 90-92.

Line 75-96. How do you measure the hearing with this approach? I can see that you quantify responses from Line 200-201, but that should be stated here, otherwise the reader is left guessing at the method. How do you justify this method, is there a study

that does so.

We have clarified the methods of testing (L 78-104) and analysis L 166-172

We have addressed concerns about behavioral testing above. L 75-78 we add in information about behavioral audiograms.

Line 100. Gras should be GRAS and you should state size (1/4", 1/8"?) and model.

Yes, I have added this information on L89.

Line 122. Did you not calibrate the Knowles microphones? Given your method, I appreciate that you do not need an absolute calibration of the entire frequency response, but a simple relative calibration should be performed (exposing both microphones to the same sound) to test how similar they are.

Yes, we compared the signals of the microphone in the setup in response to artificial clicks. The microphones were placed in the same position each time. The gain difference between them was 0.17 V and I have added this information on L 152. We have adjusted the amplitude measurements from the weaker channel accordingly and recalculated the transitions, which did not affect the overall result (L 218-219).

Line 121-134. I find this method highly inaccurate and artificial. You are holding the animal in your hand, why not place it on a platform or hang it from a mesh?

It was important in this study to get very clear recordings of such hyper-short clicks, and enough of them. In the hand, we could gently stimulate the pup and get good recordings in the direction of the microphone in a reasonable amount of time (which is important in a species in which the newborn is always attached to the mother). While artificial, this method has been used previously in other studies (e.g. Sterbing 2002).

Line 171-173. I do not understand this sentence.

This sentence was an error, thank you for noting. We have corrected and rewritten it L 208-209.

Line 180-182. You mention an "automated measurement feature" – you should either explain this in detail or post a reference that does so.

We have added in additional information on Batalef L215-217.

Line 187-193. You compare pup beam-alternations to adult beam-alternations using an intensity ratio. Given the difference in measurement scenario and low control you have over aim (hand-held pups compared to perching adults and only two microphones for animals that can vary the beam aim/beam separation) this method seems highly inaccurate. I agree that you can test if the left-left – right-right pattern is present, but comparing angles is in my opinion too uncertain with this method.

We see your concerns and removed this part.

Line 196-206. This method is not justified in your description. Interpreting calling response as validation of hearing is not entirely unjustified, but it does not guarantee it either. It is also very questionable to test the ontogeny of echolocation without exposing the animals to echolocation sounds. While you may be covering the frequency range,

the call duration is dramatically different. Again, an appropriate method would be ABR instead.

We have addressed the concerns about hearing tests from a behavioral standpoint and from ABRs above.

Following the reviewer's concerns, we used ABR's in addition to the behavioral tests. The standard ABR setup does not allow playing back an exact Rousettus click, so we have played back a click similar in duration (L 115). We also tested the response to pure tones, so altogether now demonstrate sensitivity to both echolocation click frequencies and duration. It will be an exciting advancement of this research to play back various types of stimuli (communication, echolocation, echolocation from different angles, etc) with a specialized setup in the future.

To add more information regarding the abilities of the sensory system at Day 0, we added the assessment of use of echolocation in response to sound stimuli on Day 0, showing that they emit echolocation spontaneously and at greater rates in response to a stimulus. This tells us that there is a functional basis of circuitry between sound reception with echolocation production. We also played back echolocation clicks in preliminary test to a week old pup and found she responded to the passes with isolation calls and clicks (Supplementary Fig. 3).

Line 246-257. It would be very informative to see a spectrum of the signals. While your argument concerning fft-points from line 156-157 is somewhat valid, there is a vast selection of toothed-whale literature on this that provides justification and the means to do so.

We have included spectra examples of Day 0 and Day 35 pups, and an adult (Fig. 2) Yes the toothed-whale literature does provide justification, however, this approach is a very straightforward and accurate method to acquire the peak frequency. We have referenced work by Boonman et al 2020, in which the authors compared the instantaneous and FFT frequency measurements and confirmed the correlation between these two approaches. I have added this on Line 193-195.

Referee: 2

Comments to the Author(s)

The ms by Smarsh and Yovel describes ontogenetic development of auditory responsiveness and of echolocation calls in the bat Rousettus. In most laryngeal bats, call production can be delayed after birth and the emission of adult-like calls often is coupled to the onset of flight. Rousettus, however, uses a simpler lingual echolocation production system that is based on tongue clicking. The authors show that right after

birth, *Rousettus* echolocates with clicks and the clicks are quite adult like in terms of their pairwise production. In addition, the neonate bats emit left/right click amplitude patterns that are similar to beam steering of adults. In the next weeks of life, the interpair intervals grow successively shorter and the call duration increases while the instantaneous frequency is reduced somewhat. Those changes, however, are small, compared to more massive changes during the ontogeny of laryngeal bats. In addition, the authors showed that the young bats increase the number of their own emitted isolation calls when louder sounds like echolocation clicks are presented. This of course is not a demonstration of intact hearing, but at least shows that the bats are reactive to some sounds right after birth.

The short ms generally is written well and the methods are sound. The data are important when assessing evolution of echolocation and comparing laryngeal versus lingual sound production and therefore should be published in Proc B. Lingual sound production and a concomitant perception of the clicks seems to be an innate feature and is less shaped by experience, inner ear maturation, and sensitive periods, as it is the case in laryngeal sound production.

Thank you for the careful consideration of our manuscript and comments. We have sought to address them thoroughly below.

Specific comments

Title:

your study does not demonstrate intact adult-like hearing, you did not show hearing threshold curves. But you showed that neonates did respond to echolocation signals, please change the title accordingly.

True, we have adjusted the title, and we have recorded ABRs for several day 0 – day 1 pups to add more information on the sensitivity to frequencies 6, 12, 18, 24, 30, and 35 kHz, which encompasses the most sensitive region of the *R. aegyptiacus* audiogram (Koay et al 1998). (L 115-128, 173-181, 242-250, Fig 1)

Introduction (and discussion):

I am a little bit surprised that you are not aware of a series of relevant literature with ample data on postnatal maturation of echolocation calls and corresponding behavioural adaptations in laryngeal bats (Vater M Kössl M Foeller E Coro F Mora E Russell I (2003) Development of Echolocation Calls in the Mustached Bat, *Pteronotus parnellii*. J Neurophysiol 90: 2274–2290) as well as their corresponding cochlear (Kössl M Foeller E Drexl M Vater M Mora E Coro E Russell IJ (2003). Postnatal development of cochlear function in the mustached bat, *Pteronotus parnellii*. J Neurophysiol 90:2261-2273) and auditory system maturation (Vater M Foeller E Mora EC Coro F Russell IJ Kössl M (2010) Postnatal Maturation of Primary Auditory Cortex in the Mustached Bat, *Pteronotus parnellii* J Neurophysiol 103: 2339–2354). Those data very nicely show that the mustached bat, as laryngeal bat, takes long time for maturation and would strongly emphasize the point that you make in your ms. Please cite and discuss those data.

There is an extensive amount of echolocation ontogeny literature to consider. Thank you for the suggestion, as the research on the development of the auditory system in

Pteronotus is very interesting and informative. I have referred to it in the introduction (L24, 28) and the discussion (L 306, 308-310, 320-321).

Experimental protocol:

Hearing test: To use echolocation signals for your auditory responsiveness test makes sense, but what was the rationale of using 5-110 kHz noise which probably extends beyond the auditory range of those bats. Are there hearing threshold curves obtained with pure tones available for young *Rousettus* (maybe from another publication)?

The purpose of using broad band noise was as a control. We didn't know what frequencies such a young animal would be sensitive to so we chose a broad frequency sound that would encompass the adult sensitivity range and also all of the harmonics of the communication calls we played back. We presumed they wouldn't respond to the higher frequencies of noise but also assumed it wouldn't hurt to do so.

There are no threshold curves in the literature for young *Rousettus*, so we have subsequently recorded ABR for several pups age Day 0 to 1. We reported the thresholds for the pups as well as the average for 5 adults whose ABRs were recorded in the same setup. The frequencies we used were in the main frequency sensitivity window for adult *Rousettus* (8 to 30 kHz, Koay et al 1998)(see line numbers listed in above response).

Give a little more information about the behaviour of the young bats, did they usually hang on their mothers or stayed very close? In which natural situation do they emit echolocation calls?

Thank you for noting this gap in information. We have added more information on L 70-75. Pups are always on their mothers for the first 3 weeks of age. As for echolocation, we did not have expectations or prior knowledge of echolocation use at such young ages. We observed the echolocation emission when we first started playing back adult social calls and noise at the start of the project. We have gone back through the data and observed that pups untouched in the playback experiment will emit echolocation click pairs on their own in between isolation calls, but the acoustic stimuli significantly increase emission. We have added a video of playback to a Day 0 pup and example spectrograms of the response in the supplemental to illustrate this (Supplementary Media, Supplemental Fig 2). We have analyzed the echolocation response and included it in the methods (L 80-104, L166-172), and results (L234-241), and in the figures (Fig. 1). This provides greater input in hypotheses regarding the development of echolocation vs communication from a neural standpoint. We have only found one study (Esser & Daucher 1996 Hearing in the FM-bat *Phyllostomus discolor*: a behavioral audiogram) in the literature in which interactive playbacks of pups were conducted in neonates. The response to playbacks of stimuli in our study illustrates that there is a functional basis of the auditory and vocal motor circuitry at this age, and also leads to further research questions regarding differentiation of sensorimotor circuitry at this age for laryngeal sounds vs tongue clicks.

Echolocation signal collection: what was the sound pressure level of the echolocation calls of the young bats? Since their tongues still undergo anatomical maturation, were

the signal levels lower than those from adults?

We have gone back and calibrated the recording the settings and estimated the source levels (range 94.8 to 107 db SPL 10 cm from the mouth). We have added it here, L 286-287. We couldn't compare these values to our adult recordings, but the pup values were not as high as the adult intensities reported in the literature: 75-125 db SPL or 105-115 db SPL at 10 cm from the mouth measured in the lab (reviewed in Yovel et al 2011).

Hearing assessment: When you acoustically stimulated the bats and they produced more isolation calls, they also emitted echolocation calls in this situation. Was their number also increased?

Interesting question. We went back and analyzed the trials for the same individuals from 2018, and included the data as stated above. They did respond to stimuli with more echolocation (L234-241, Fig. 1). We also included spectrograms showing vocal responses to echolocation passes in a young pup in the Supplemental material (Supp. Fig. 3).

Results

L199: detachment of pups from mothers also happens naturally? Stay the pups at the roost while the mothers fly out to feed?

They are on the mother continuously for the first 3 weeks, then the mothers start leaving the pups behind while foraging. We have added this information L 72-73.

Discussion

I.278 I am not quite sure how to interpret your statement about cochlear growth rate. Do you want to say that cochlear development in lingual bats is finished earlier than cochlear development in laryngeal bats (which would support your suggestion that hearing also matures faster)? I would not rely too much on macroscopic cochlea growth data since there is an extended period of maturation of cochlear function after the cochlea has reached its final size. In this respect, microstructural changes of basilar membrane and tectorial membrane stiffness and coupling between hair cells and those membranes are important and will increase cochlear sensitivity in a frequency specific way in young bats. Could you give more information what you are referring to here?

We originally stated this as evidence strengthening the hypothesis that laryngeal and lingual echolocators had a common echolocating ancestor, but following the reviewer's comment we have reviewed and reformulated this section in the discussion. In lingual bats the majority of external cochlear development occurs prenatally, followed by a slow trajectory postnatally, rather than what may be continual rapid changes in laryngeal bats. This points to different ontogenetic and perhaps evolutionary patterns. Furthermore, we may hypothesize that multiple aspects of the sensory system are in place at birth in lingual bats, with additional, *yet minimal*, refinements occurring slowly compared to laryngeal bats (which have dramatic shifts in frequency corresponding to many changes in the hair cells, cochlear neurons, and auditory cortex). Flight occurs much later in *Rousettus* as well. We have rewritten this section and our emphasis L 307-319, 320-324. We welcome your input on these ideas.

Journal Code: RSPB

Print ISSN: 0962-8452

Online ISSN: 1471-2954

Journal Admin Email: proceedingsb@royalsociety.org

MS Reference Number: RSPB-2020-2971

Article Status: REJECTED

MS Dryad ID: RSPB-2020-2971

MS Title: Intact hearing, echolocation, and beam steering from Day 0 in tongue-clicking bats

MS Authors: Smarsh, Grace; Yovel, Yossi

Contact Author: Grace Smarsh

Contact Author Email: gcsmarsh@gmail.com

Contact Author Address 1:

Contact Author Address 2:

Contact Author Address 3:

Contact Author City: Rehovot

Contact Author State:

Contact Author Country: Israel

Contact Author ZIP/Postal Code: 69978

Keywords: lingual echolocation, *Rousettus aegyptiacus*, ontogeny, active sensing, pup behavior

Abstract: Bats of the genus *Rousettus* are the only animals known to naturally produce click echolocation from the tongue. There have been many studies examining the dramatic changes of laryngeal echolocation in pups across bat families, but little is known about the ontogeny of lingual echolocation. Here, we examined the echolocation development of *Rousettus aegyptiacus*, the Egyptian fruit bat, which uses rapid tongue movements to produce hyper-short clicks and steer the direction of the beam. We recorded echolocation once a week from Day 0 through Day 35 postbirth and examined temporal emission patterns, signal frequency, and signal duration. We assessed the age of hearing and beam-steering abilities. On Day 0 postbirth *R. aegyptiacus* pups hear and produce hyper-short clicks in a paired pattern, the same as adults. Remarkably, newborn pups were able to use their tongues to steer the sonar beam, showing that *Rousettus* pups are born with highly developed tongue control. As they aged, pups produced click pairs faster, converging with adult intervals by age of first flights (7-8 weeks). In contrast to laryngeal bats, *Rousettus* echolocation frequency and duration is stable through Day 35, but shift by the time pups begin to fly, possibly due to tongue-diet maturation effects. Furthermore, frequency and duration shift in the opposite direction of mammalian laryngeal vocalizations. *Rousettus* lingual echolocation thus appears to be a highly functional, intact sensory system from birth. This nearly-innate sensing might come at the cost of behavioral flexibility that allows laryngeal bats to adapt their calls to different environments.

EndDryadContent

Appendix B

Author response to second round of reviews for “Hearing, echolocation, and beam steering from Day 0 in tongue-clicking bats:”

Associate Editor

Board Member

Comments to Author:

Thank you for your careful revisions of this manuscript.

Reviewer(s)' Comments to Author:

Referee: 1

Comments to the Author(s).

The authors have addressed all my concerns in a very nice way and I have no further comments. I am happy to recommend the study for publication

Dear Reviewer(s),

We were happy to see that our efforts to respond thoroughly to the Reviewers and the Assoc. Editor sufficiently addressed the concerns on our manuscript.

Thank you for the careful reviews and comments throughout this process. We think that the manuscript and overall story has been greatly improved.

Kind regards,

Authors